# Pathogen and human NDPK-proteins promote AML cell survival via monocyte NLRP3-inflammasome activation

Sandro Trova[1], Fei Lin[1], Santosh Lomada[2], Matthew Fenton[1], Bhavini Chauhan[1], Alexandra Adams[1], Avani Puri[1], Alessandro Di Maio[1], Thomas Wieland[2], Daniel Sewell[1], Kirstin Dick[1], Daniel Wiseman[3], Deepti P. Wilks[4], Margaret Goodall[5†], Mark T. Drayson[5], Farhat L. Khanim[6☉], Christopher M. Bunce[1☉]*

1 School of Biosciences, University of Birmingham, Birmingham, United Kingdom, 2 Institute of Experimental and Clinical Pharmacology and Toxicology, Heidelberg University, Mannheim, Germany, 3 Division of Cancer Sciences, University of Manchester, Manchester, United Kingdom, 4 Cancer Research UK Manchester Institute, Manchester Cancer Research Centre Biobank, The University of Manchester, Manchester, United Kingdom, 5 Institute of Immunology and Immunotherapy, University of Birmingham, Birmingham, United Kingdom, 6 Clinical Sciences, University of Birmingham, Birmingham, United Kingdom

☉ These authors contributed equally to this work.
† Deceased.
* c.m.bunce@bham.ac.uk

**Data Availability Statement:** All relevant data are within the paper and its Supporting Information files.

## Abstract

A history of infection has been linked with increased risk of acute myeloid leukaemia (AML) and related myelodysplastic syndromes (MDS). Furthermore, AML and MDS patients suffer frequent infections because of disease-related impaired immunity. However, the role of infections in the development and progression of AML and MDS remains poorly understood. We and others previously demonstrated that the human nucleoside diphosphate kinase (NDPK) NM23-H1 protein promotes AML blast cell survival by inducing secretion of IL-1β from accessory cells. NDPKs are an evolutionary highly conserved protein family and pathogenic bacteria secrete NDPKs that regulate virulence and host-pathogen interactions. Here, we demonstrate the presence of IgM antibodies against a broad range of pathogen NDPKs and more selective IgG antibody activity against pathogen NDPKs in the blood of AML patients and normal donors, demonstrating that *in vivo* exposure to NDPKs likely occurs. We also show that pathogen derived NDPK-proteins faithfully mimic the catalytically independent pro-survival activity of NM23-H1 against primary AML cells. Flow cytometry identified that pathogen and human NDPKs selectively bind to monocytes in peripheral blood. We therefore used vitamin D₃ differentiated monocytes from wild type and genetically modified THP1 cells as a model to demonstrate that NDPK-mediated IL-1β secretion by monocytes is NLRP3-inflammasome and caspase 1 dependent, but independent of TLR4 signaling. Monocyte stimulation by NDPKs also resulted in activation of NF-κB and IRF pathways but did not include the formation of pyroptosomes or result in pyroptotic cell death which are pivotal features of canonical NLRP3 inflammasome activation. In the context of the growing importance of the NLRP3 inflammasome and IL-1β in AML and MDS, our findings now implicate pathogen NDPKs in the pathogenesis of these diseases.

**Funding:** This research was supported by a grant from Blood Cancer UK (Trading name of Bloodwise) Grant 17011) CB, FK (https://bloodcancer.org.uk/). SL has been supported by the state of Baden-Württemberg within the PharmCompNet project. The funders had no role in study design, data collection and analysis, decision to publish, or preparation of the manuscript.

**Competing interests:** The authors have declared that no competing interests exist.

## Introduction

Population based studies have identified that a history of infection associates with increased risk of both acute myeloid leukaemia (AML) and the related myelodysplastic syndromes (MDS) [1] which also have a high risk of progression to AML. Infection related risk remains significant even when confined to infections occurring three or more years before AML/MDS diagnosis [1]. Neutropenic sepsis is also the most common route to AML presentation and a major cause of death in both AML and MDS [2–5]. Following diagnosis, infections continue to represent key clinical challenges in AML/MDS treatment and management. It is generally accepted that increasing immune impairment arising as a consequence of disease progression is responsible for the frequency of life-threatening infections in these patients. However, the reciprocal role of infections in the development and progression of AML and MDS remains poorly understood and largely ignored.

Despite this, the important role of the inflammatory cytokine interleukin-1β (IL-1β) in AML has become increasingly recognized. Early studies demonstrated that soluble IL-1 receptor (sIL-1R) and IL-1β receptor antagonists (IL-1RA) reversed the pro-survival effect of IL-1β on AML blasts *in vitro* [6]. More recently, a functional screen of 94 cytokines identified that IL-1β elicited expansion of myeloid progenitors whilst suppressing the growth of normal progenitors in 67% of AML patients [7]. In the same study, silencing of the IL-1 receptor led to significant suppression of clonogenicity and *in vivo* disease progression [7]. Similarly chronic IL-1β exposure triggers the selective *in vivo* expansion of *Cebpa*-deficient multipotent hematopoietic progenitors, demonstrating a role for inflammation in selecting early premalignant clones in the bone marrow [8]. A recent study also identified that AML patients with low IL-1β along with high levels of the IL-1 receptor antagonist IL-1RA were protected against relapse following immunotherapy, further implicating inflammation in promoting relapse [9].

IL-1β secretion is regulated by activation of multiprotein complexes called inflammasomes. The *IL1B* gene encodes an inactive cytoplasmic pro-IL-1β protein that is cleaved upon inflammasome activation by caspase-1, to form the active secreted from of IL-1β. Several studies have implicated the NLRP3-inflammasome in AML progression [10–12]. Equally, the importance of NLRP3 activation in the pathogenesis of myelodysplastic syndromes (MDS) is well established [13–16]. The NLRP3 inflammasome (comprised of the NOD-, leucine-rich repeat (LRR)-and pyrin domain (PYD)-containing protein 3 (NLRP3), the adapter apoptosis-associated speck-like protein containing a caspase recruitment domain (ASC), and the effector protease caspase-1) orchestrates Toll-like receptor 4 (TLR4) responses to both sterile and infection-related inflammation. Sterile inflammation occurs in response to host derived damage associated molecular pattern (DAMP) signals (including S100A9 (S100 Calcium Binding Protein A9)) [13,17], whereas infection-related inflammation is mediated by pathogen associated molecular pattern (PAMP) signals (classically LPS (lipopolysaccharide) from Gram-negative bacteria) [18,19].

Following on from reports that elevated plasma levels of the human nucleoside diphosphate kinase (NDPK) NM23-H1 are associated with poor prognosis in AML [20–22], we and others demonstrated that exogenous NM23-H1 indirectly promotes AML blast cell survival via the induction of inflammatory cytokines including IL-β [23–25]. NDPK proteins (also termed NDK, NME) are evolutionarily highly conserved across prokaryotes and eukaryotes. The most studied function of NDPKs is catalysis of the reversible γ-phosphate transfer from nucleoside triphosphates (NTPs) to NDPs [26]. However, microbial NDPKs also have complex roles in virulence [27,28], regulating immune responses to infection [28,29] and host-pathogen interactions [30–32]. Therefore, it is now important to consider whether, in addition to NM23-H1 driving AML progression, infection derived NDPKs may also directly exacerbate risk of AML and MDS and disease progression.

We demonstrate here that NDPKs from pathogens commonly associated with infections and sepsis in AML and MDS patients, faithfully recapitulate the activity of NM23-H1 in promoting primary AML cell survival. We further demonstrate that human and pathogen NDPKs induce IL-1β production in monocytes via an NLRP3 and caspase-1 dependent but TLR4-independent mechanism. Monocyte stimulation by NDPKs also resulted in activation of NF-κB and interferon regulatory factor (IRF) signaling pathways, but did not include the formation of pyroptosomes or pyroptotic cell death, which are pivotal features of canonical NLRP3 inflammasome activation [33,34]. Thus, our data indicate a novel role for pathogen derived NDPK proteins in promoting disease progression in AML and MDS.

## Materials and methods

### Ethics

Primary samples were obtained with written informed consent under ethical approval: University of Birmingham, ERN_17–0065 for healthy blood samples and by NRES Committee North West—Cheshire REC reference 12/NW/0742 for AML bone marrow aspirates.

### Cell lines and tissue culture

THP-1 (DSMZ) and THP-1 NLRP3 deficient ($N_{def}$), THP-1 Caspase-1 deficient ($Casp1_{def}$), THP-1 NLRP3 knockout ($N_{ko}$), THP-1 ASC::GFP, THP1-Dual, and THP1-Dual-TLR4 KO Cells (all Invivogen) were maintained between $0.5-1x10^6$ cells/ml in RPMI 1640 media, supplemented with 10% Fetal Bovine Serum and 100U/ml penicillin, 100μg/ml streptomycin (pen/strep) (complete media) at 37˚C, 5% $CO_2$ in a humidified incubator. To select for reporter gene expression THP1-Dual and THP1-Dual TLR4 KO cultures were also supplemented with 100μg/ml Zeocin and 10μg/ml Balsticidin. Cells were STR-profiled regularly throughout the study.

### DR THP1 and TLR4 KO DR THP-1 Cell reporter assays

THP1-Dual™ and TLR4 KO THP1-Dual™ cells feature the secreted Lucia luciferase reporter gene, under the control of an ISG54 (interferon-stimulated gene) minimal promoter in conjunction with five interferon (IFN)-stimulated response elements. The same cells also express a secreted embryonic alkaline phosphatase (SEAP) reporter gene driven by an IFN-β minimal promoter fused to five copies of the NF-κB consensus transcriptional response element and three copies of the c-Rel binding site. These reporter proteins were measured in cell culture supernatants using QUANTI-Luc™ (IRF activity) and QUANTI-Blue™ (NF-κB activity) assays as described by the manufacturer (Invivogen).

### Protein production

The complete coding sequences of NM23-H1, bacterial and fungal NDPKs were cloned into a pET15b backbone (Merck Millipore) and transformed into BL21 (DE3) or ClearColi BL21 (DE3) bacteria. Protein expression was induced with 1mM Isopropyl β-D-1-thiogalactopyranoside (IPTG) overnight at 20˚C in a shaking incubator. Recombinant proteins were purified with His-Bind Resin and His-Bind Buffers (Merck Millipore) according to manufacturer's protocol and stored at a concentration of 2mg/ml in elution buffer (Merck Millipore) at 4˚C or diluted to 0.2mg/ml in culture media and filter sterilized for cell treatments.

## Indirect ELISA for anti-NDPK IgG and IgM

Nunc MaxiSorp flat-bottom ELISA Plates (ThermoFisher Scientific) were coated with 100μl of rNDPK at 1μg/ml in PBS overnight at 4˚C or PBS alone. Plates were then washed for 4 times with 300μl of PBS-Tween-20 (0.2% Tween-20, PBS-T) and blocked with 200μl of 2% BSA in PBS for 1 hour at room temperature, shaking at 500rpm. Plates were then washed once with PBS-T and incubated with 100μl of patient plasma/serum diluted 1:200 in PBS-T for 1 hour at room temperature whilst shaking. Plates were washed 4 times with PBS-T and incubated with 100μl of HRP-labelled specific α-human IgG (Clone R10) and α-human IgM (Clone AF6) antibodies diluted 1:1000 in PBS for 1 hour at room temperature whilst shaking. Plates were washed 5 times with PBS-T and 100μl of 3,3′,5,5′-Tetramethylbenzidine (TMB) (Merck Millipore) equilibrated at room temperature were added, and reaction was blocked after 15 minutes with 2N $H_2SO_4$. Plates were read at 450nm with VICTOR X3 Light Plate Reader (PerkinElmer). Absorbance was then calculated by subtracting the cross-reactivity of HRP-labelled detection antibodies against rNDPK and baseline obtained for blocking solution.

## Primary AML direct survival assay

Bone marrow leukocytes were washed in RPMI 1640 supplemented with pen/strep (Gibco), and resuspended in RPMI 1640 supplemented with 1% v/v ITS+ Premix (Corning) and pen/strep (Serum-free media) at $1x10^6$ cells/ml. Treatments were performed with 2μg/ml of rNM23-H1/rNDPKs, in the presence of 1.25μg/ml polymyxin B (Merck Millipore) (see also S1 Fig). Cells were incubated for 7 days and immunostained for flow cytometry (Antibodies: CD34, clone 581; CD117, clone: 104D2; CD11b, clone ICRF44 (BD Pharmingen) and 1:50 FcR Blocking Reagent (Miltenyi Biotec)). Cells were analyzed on a FACS Calibur (BD Pharmingen) in the presence of counting beads for live cells enumeration.

## Cell cycle analysis

Cell cycle was assessed by incubating 300μl of cells with 200μl of Cell Cycle Buffer (30μg/mL Propidium Iodide (PI), 0.1mM NaCl; 1% Triton X-100) for 24 hours at 4˚C in the dark and before analysis on a BD FACS Calibur.

## rNDPK Alexa 647 labelling

Purified rNM23-H1, rNDPK form *S.pneumoniae* and Bovine serum albumin (BSA) were diluted to 2mg/ml and dialyzed at 4˚C overnight in 2000 volumes of 0.5M NaCl, 0.1M imidazole in phosphate buffered saline (PBS), pH 8.2, prior to labelling using the Alexa Fluor 647 Protein Labelling Kit (ThermoFisher Scientific) according to manufacturer's instructions.

## rNDPK Binding to normal leukocytes

Normal adult donor blood (in EDTA) was diluted 1:5 with RPMI 1640 supplemented with pen/strep and 2mM L-glutamine (Gibco). Cells were incubated at 37˚C for 2 hours in the presence of polymyxin B (1.25μg/ml) and either 2μg/ml of Alexa 647 labelled NDPKs or BSA. Red cells were then lysed with red cell lysis buffer (155mM ammonium chloride, 10mM potassium bicarbonate, 0.1mM EDTA, pH 8.0) and leukocytes immunostained for CD19 (clone SJ25C1) CD3 (clone SK7), CD11b (clone D12), and CD14 (clone M5E2) (all BD Pharmingen), in the presence of 1:50 FcR Block (Miltenyi Biotec) before flow cytometric analysis.

## Differentiation of THP-1 monocyte models

Prior to treatments, THP-1 cell lines were diluted to $0.2 \times 10^6$ cells/ml and differentiated with 100nM $1\alpha$,25-dihydroxy Vitamin $D_3$ (Cayman Chemical) for 72 hours. Differentiation was confirmed by measuring expression of CD11b and CD14 markers by flow cytometry (antibodies: CD11b, clone D12; CD14, clone M5E2, BD Pharmingen).

## Normal donor mononuclear preparation

Peripheral Blood Mononuclear Cells (PBMC) from normal adult donor blood were isolated using Ficoll-Paque PLUS (Fisher Scientific). After purification, cells were resuspended at $1 \times 10^6$ cells/ml in 1% ITS+ RPMI 1640 media for the treatments.

## Monocyte depletion

Red cells from normal adult donor blood (in EDTA) were lysed in red cell lysis buffer and leukocytes were rested for 24 hours in a non-coated flask in RPMI 1640 media supplemented 10% autologous plasma. After 24 hours, total leukocytes were depleted from monocytes with CD14 MicroBeads UltraPure (Miltenyi Biotec) according to manufacturer's instruction and treated in autologous media.

## Cytokine analysis

Diluted whole blood (1:5), $1 \times 10^6$ PBMC/ml, total leukocytes, monocyte-depleted leucocytes, or Vitamin $D_3$ differentiated THP-1/ml were treated with 2µg/ml of rNM23-H1 or rNDPKs, and when possible with 2µg/ml of ultrapure lipopolysaccharide from *E.coli* K12 (Invivogen) and 20µM Nigericin (Merck Millipore) as control for alternative and canonical inflammasome, respectively. Polymyxin B (PMB) 1.25µg/ml was used for all the treatments except LPS. Cells were incubated for 18 hours and conditioned media (CM) collected by centrifugation and stored at -20°C. IL-1β and IL-6 were analyzed by ELISA MAX Deluxe Sets (Biolegend) according to manufacturer's instruction.

## Caspase-1 activation

**Normal leukocytes.**   Normal adult donor blood (EDTA) was diluted 2 + 3 in RPMI160. Cells were treated as for the cytokine analysis for one hour with Nigericin, LPS, rNM23-H1 and rNDPK and incubated at 37°C. FLICA 660 Caspase-1 (Bio-Rad) substrate was then resuspended in DMSO following the manufacturer's protocol and applied 1:30 on the cells. Non-FLICA treated cells were vehicle controlled. Cells were then incubated for a further 1.5 hours with gentle pipetting every 30 minutes. Red cells were then lysed and leukocytes immunophenotyped by flow cytometry analysis as described above. Caspase-1 positive cells were calculated with Population Comparison (SE Dymax) in FlowJo 10.6. Version 10.6.0.

**THP-1 cell lines.**   Cells were plated and treated as for the cytokine analysis, but incubated for 2.5 hours. CM was then harvested by centrifugation and active caspase-1 measured with the Caspase-Glo 1 Inflammasome Assay (Promega) following the manufacturer's instructions. Luminescence was measured after 90' with VICTOR X3 Light Plate Reader (PerkinElmer). Luminescence values obtained in the presence of Ac-YVAD-cho inhibitor were subtracted from the non-inhibited reaction, to obtain the caspase-1-specific activation.

## Inhibition of inflammasome components

Inhibitors and antagonists used were: Caspase-1 inhibitor Ac-YVAD-cmk (Invivogen), 18.5µM in DMSO; NLRP3 inhibitor MCC950, 10µM in DMSO; TLR4 inhibitor TAK242

2.76μM in DMSO. Cells were pre-treated for 1 hour with the inhibitors, while TLR4 antagonist was added from the beginning of the incubation period.

## ASC Speck/Pyroptosome formation

Pyroptosome formation (ASC speck) was investigated both with THP-1 ASC::GFP cell line (Invivogen) and THP-1 wild type. Cells were differentiated with Vitamin $D_3$ prior to treatment. THP-1 ASC::GFP nuclei were stained with 250ng/ml Hoescht 33342 (Merck Millipore) for 45 minutes, washed and resuspended at $0.5x10^6$ cells/ml in phenol red free complete media and treated with 2μg/ml of LPS or rNDPK. For nigericin treatment, since it cannot induce the NfκB-dependent fusion gene ASC::GFP, it was necessary to pre-treat cells with 2μg/ml LPS and then with 20μM Nigericin (LPS+Nig) in order to induce canonical inflammasome and visualize pyroptosomes (ASC specks). THP-1 wild type were used to immunostain for ASC specks. Cells were treated for 6 hours and cytospun on glass slides. After fixation with 4% paraformaldehyde, cytospins were blocked with 10% heat inactivated goat serum (Merck Millipore), 1% FBS (Gibco) in PBS (Gibco) for 30 minutes at 37˚C and then incubated with 10 μg/ml of αASC TMS-1 (clone HASC-71, Biolegend) in blocking buffer, overnight, at 4˚C. Texas Red labelled secondary antibody (Jackson ImmunoResearch) was used at 1:200, for 1 hour, at 37˚C. Micrographs were acquired with EVOS FL (ThermoFisher Scientific) and analysis was performed with the ImageJ distribution [35].

## Primary AML indirect survival assay

CM was prepared from normal donor 1:5 diluted whole blood or $1x10^6$ PBMC/ml +/- rNDPK and rNDPK depleted as described before [24]. AML cells from 3 primary AML bone marrow aspirates were resuspended in CM at $1x10^6$ cells/ml, incubated for 7 days and viability assessed with flow cytometry and counting beads.

## Quantification and statistical analysis

Results are expressed as means ± standard error (SEM) from at least three independent replicates for each experimental group. Statistical analyses were performed with GraphPad Prism 9 and the relevant test used is specified in the Figure legends. Values of $p < 0.05$ are identified with an *.

# Results

## NM23-H1 and pathogen NDPKs are highly conserved

We aligned the human NM23-H1 amino acid sequence with publicly available amino acid sequence data for NDPKs from four bacterial strains and one fungal strain commonly associated with infections in AML and MDS patients, namely, *Escherichia coli (E.coli)*, *Staphylococcus aureus (S.aureus)*, *Streptococcus pneumoniae (S.pneumoniae)*, *Klebsiella pneumoniae (K. pneumoniae)*, and *Candida albicans C.albicans* (Fig 1A). Sequence conservation varied from 42–64% (S1 Table) and included conservation of key residues (red stars) required for enzymatic and oligomerization functions in NM23-H1 (Fig 1A). We also used available crystal structures to compare the tertiary structures of NM23-H1 with NDPKs from *S.aureus and E. coli* and the quaternary structures of NM23-H1 and *S.aureus* hexameric complexes (Fig 1B). Together, these analyses reveal remarkable conservation of human, prokaryotic and yeast pathogen NDPK proteins across millennia of evolution.

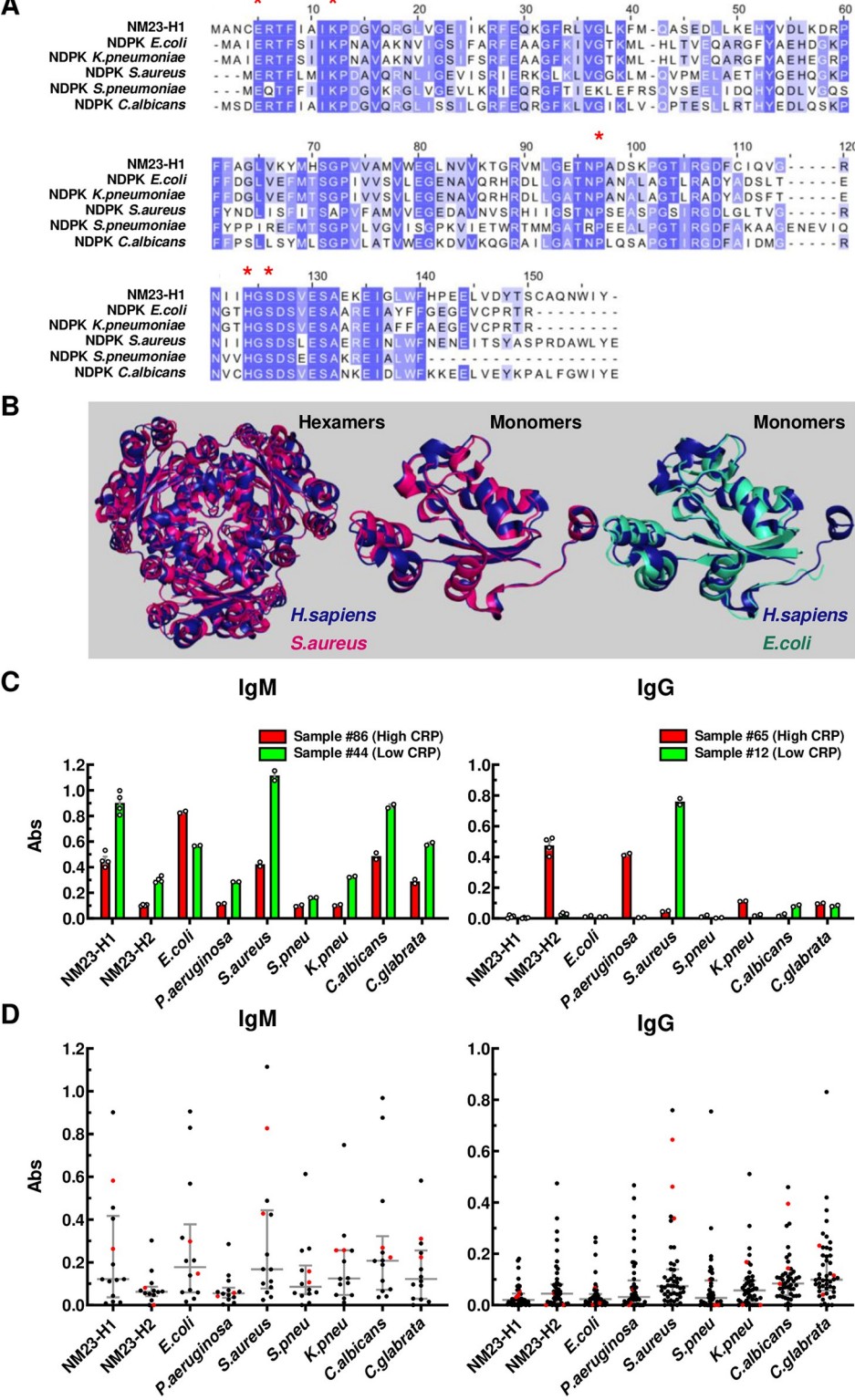

**Fig 1. NM23-H1 and pathogen NDPKs are highly conserved and recognized by human humoral antibody responses.** (A) NDPK protein sequences alignments were obtained with ClustalW and visualised with JalView by Percentage Identity. Red * indicates the key residues for NM23-H1 enzymatic activities and structural organization. (B) 3D alignment of NM23-H1 (blue) crystal structure (pdb 1JXV) to *S.aureus* NDPK (red) tetramer and monomer (pdb 3Q83) and *E.coli* (green) monomer (pdb 2HUR) were generated with PyMol 2.3 (See also S1 Table). (C)

Representative data of four independent samples (#86 and 44 for IgM; #65 and 12 for IgG), two with high CRP (red bars) and two with low CRP (green bars). Averages are expressed as mean ± SEM. (D) Dot plots for all the analyzed samples; black dots: AML samples; red dots: Normal donor samples. Grey bar indicates median ± interquartiles. All samples measured for IgM activity were also measured for IgG activity. CRP values and individual antibody activities for all samples are summarized in S2 Table.

## Detection of humoral immunity to pathogen NDPKs

The extensive conservation between human and pathogen NDPKs questions whether the human innate or adaptive immune system can detect and respond to them. However, evidence of humoral immunity to pathogen NDPKs would indicate that exposure and response to pathogen NDPKs occurs *in vivo*. We therefore screened sera from AML patients and normal donors for IgM- and IgG-reactivity against recombinant NDPKs (rNDPKs) from *E.coli, Pseudomonas aeroginosa (P.aueruginosa), S.aureus, S.pneumoniae, K.pneumoniae, C.albicans* and *Candida glabrata (C.glabrata)* and the human NDPKS (NM23-H1 & -H2) (Fig 1C and 1D). IgM reactivities were tested in 14 AML patients and 2 normal donors. Most donors exhibited IgM activity against a broad spectrum of pathogen and human NDPKs (Fig 1C and 1D) indicative of B1 B-cell immunity which is fetal in origin and provides natural broad affinity protective IgM anti-PAMP antibodies in the steady state in the absence of antigenic stimulation [36]. In contrast, we observed greater variation and selectivity between individuals in IgG reactivity to pathogen NDPKs. For example AML sample #12 displayed selective IgG reactivity with *S. aureus* NDPK, whereas AML sample #65 displayed selective reactivity with *P. aueruginosa* NDPK (Fig 1C). Despite this variation, across a total of 45 AML (selected as described in S2 Table) and 3 normal donor samples, we identified marked IgG responses to all pathogen NDPKs tested (Fig 1D). NM23-H2 is largely intracellular and serum levels are very low outside of disease. Therefore, these observations suggest that humans experience exposure to pathogen NDPKs *in vivo* and, in doing so, raise antibody repertoires capable of cross-reactivity with human NDPKs which are primarily intracellular during health.

## Pathogen NDPKs recapitulate the actions of NM23-H1 against primary AML cells

We investigated the ability of pathogen NDPKs to recapitulate the actions of NM23-H1 in promoting *in vitro* AML blast cell survival in a significant subset of AMLs [23,24]. We treated 16 individual primary AMLs with rNM23-H1 and pathogen rNDPKs. As shown in Table 1, samples that responded to rNM23-H1 also responded to pathogen rNDPKs, whereas a lack of NM23-H1 response associated with a lack of response to pathogen NDPKs. Seven of 16 (44%) AMLs demonstrated a mean blast cell survival index (BSI; defined as ratio of blasts in NDPK treated cultures versus untreated controls) of >1.5 compared to controls when exposed to rNM23-H1 or pathogen rNDPKs (for absolute cell numbers see S3 Table). The mean BSI in this group was 4.27 whereas in the non-responders it was 1.03 (p = 0.0047 unpaired t-test with Welch's correction / 0.0002 Mann Whitney U-test). This response rate is consistent with response rates to NM23-H1 previously published by ourselves and others [23,24].

Fig 2 shows in more detail the responses of three selected AMLs (AML1, -2, -4) that displayed the range of responses to rNM23-H1 and pathogen rNDPKs. AML1 demonstrated rNDPK-independent cell survival. This included survival of both AML blasts, identified as CD34+ and/or CD117+ cells, and of the remaining more mature CD34-/CD117- cells in the sample (Fig 2A and 2B) (See S1 Fig for flow cytometry gating strategy). In contrast, untreated AML2 cells survived poorly *in vitro*, but addition of rNM23-H1 and pathogen rNDPKs equally and markedly increased survival of blasts and more mature cells in the sample (Fig 2A and

**Table 1. NM23-H1 and pathogen NDPKs exert the same pro-survival effect on primary AML cells.**

| Sample | ctrl | H1 | EC | SP | KP | SA | CA | mean |
|---|---|---|---|---|---|---|---|---|
| | | | | Survival index | | | | |
| AML10 | 1.00 | 10.25 | 6.85 | 6.27 | nt | nt | 11.04 | 8.60 |
| AML2 | 1.00 | 7.27 | 8.55 | 5.35 | 5.16 | 7.94 | nt | 6.85 |
| AML4 | 1.00 | 6.21 | 4.91 | 6.94 | 8.20 | 7.82 | nt | 6.82 |
| AML16 | 1.00 | 3.50 | 1.93 | nt | nt | nt | 1.55 | 2.33 |
| AML11 | 1.00 | 1.59 | 1.47 | 1.89 | nt | nt | 2.47 | 1.86 |
| AML3 | 1.00 | 1.55 | 2.47 | 1.71 | 1.03 | 2.07 | nt | 1.76 |
| AML12 | 1.00 | 1.34 | 1.56 | 1.90 | nt | nt | 1.80 | 1.65 |
| AML15 | 1.00 | 1.48 | 1.23 | nt | nt | nt | nt | 1.36 |
| AML6 | 1.00 | 1.56 | 1.21 | 1.17 | 1.24 | 1.52 | nt | 1.34 |
| AML8 | 1.00 | 1.15 | 1.18 | 1.20 | 1.09 | 1.03 | nt | 1.13 |
| AML9 | 1.00 | 1.01 | 1.01 | 1.10 | 1.22 | 0.93 | nt | 1.05 |
| AML1 | 1.00 | 0.92 | 1.05 | 0.97 | 0.85 | 1.01 | nt | 0.96 |
| AML5 | 1.00 | 0.77 | 0.72 | 1.72 | 0.98 | 0.55 | nt | 0.95 |
| AML14 | 1.00 | 1.00 | 0.84 | 1.20 | nt | nt | 0.45 | 0.87 |
| AML13 | 1.00 | 0.84 | 0.72 | 0.82 | nt | nt | 0.89 | 0.82 |
| AML7 | 1.00 | 0.86 | 0.66 | 0.62 | 0.72 | 1.00 | nt | 0.77 |

Survival indices (SI) are shown for 16 primary AML samples (AML1-16) in response to rNDPKS from Human (H1), *E. coli* (EC), S.pneumoniae (SP), K.pneumoniae (KP) S.aureus (SA) and C.albicans (CA). Samples are ranked highest to lowest with respect to the mean (SI). A mean SI of greater than 1.5 was deemed as a positive response (samples highlighted in blue).

2B). Survival of AML4 in the absence of rNM23-H1 or pathogen rNDPKs was similar to AML1 (Fig 2A and 2B). However, unlike in AML1, both rNM23-H1 and pathogen rNDPKs equally further increased survival of AML4 blast and mature cells (Fig 2A and 2B). In this sample, both rNM23-H1 and pathogen rNDPKs also supported some AML cell proliferation as measured by cells in S+$G_2$M phase in flow cytometric cell cycle analyses (Fig 2C). Combined, the data in Table 1 and Fig 2 demonstrate that AML cell responses to pathogen NDPKs are indistinguishable from the previously reported responses to NM23-H1 [23,24].

## Promotion of AML cell survival by both human and pathogen rNDPKs is mediated via interaction with non-malignant myeloid cells

We used fluorescently labelled rNM23-H1 and *S.pneumoniae* rNDPK to test for binding to normal donor peripheral blood cells. Both rNM23-H1 and *S.pneumoniae* rNDPK proteins bound to monocytes but not B-cells, T-cells or neutrophils (Fig 3A and 3B). Notably binding of *S.pneumoniae* NDPK to monocytes appeared less strong than NM23-H1 (Fig 3B, note log scale). We next measured IL-1β and IL-6 production by normal donor peripheral blood cells when exposed to rNM23-H1, *S.pneumoniae* rNDPK and *C.albicans* rNDPK. All three NDPKs induced IL-1β and IL-6 production in PBMC cultures (Fig 3C). However, production of both cytokines was enhanced in cultures of total white cells (TWCs) (Fig 3C) indicating that, although the initial response requires binding of NDPKs to monocytes, the presence of neutrophils in TWC acts to amplify the cytokine response. Whereas at diagnosis AML patients are severely neutropenic, the role of neutrophils in infections during remission might be important. Furthermore, many MDS patients with high risks of AML progression have reasonable numbers of neutrophils that could be involved in pathogen NDPK responses. To further interrogate the importance of monocytes, we exposed leukocytes after

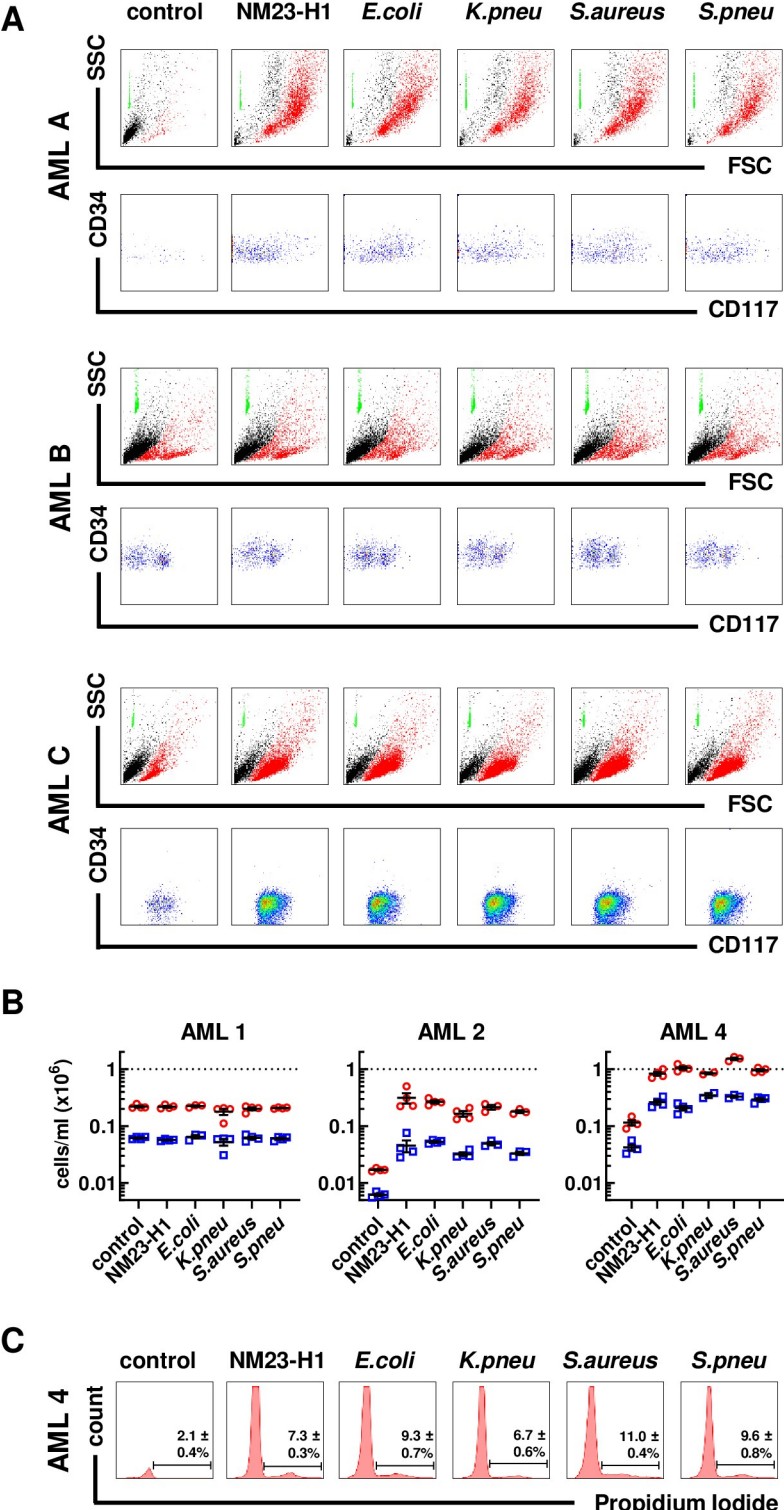

**Fig 2. Pathogen NDPKs mimic the action of NM23-H1 against primary AML cells.** (A) Representative flow cytometer plots for three primary AML cells treated with rNDPKs for 7 days. Live (in red) and dead cells (in black) were identified by Forward Scatter (FSC) and Side Scatter (SSC). Counting beads (in green) were used to enumerate live cells. Immunostaining with CD34 and CD117 identified the AML blasts amongst live cells. (B) Dot plots show the mean ± SEM of at least 3 replicates for live cells/ml (red circles) and live blasts/ml (blue squares). Dotted line is number of total viable cells at day 0. (C) Flow cytometry histograms of AML 4 cells stained with propidium iodide for cell cycle

analysis. Bars and percentages represent cells in $S/G_2M$ phase. AML 2 and AML 4 treatments are all statistically different to the controls; $p<0.05$, One-way ANOVA with Dunnet's test. Gating strategy is presented in S1 Fig.

red cell lysis to NM23-H1, *E.coli* and *C.albicans* rNDPKs before and after monocyte depletion and measured the resultant secretion of IL-1β and IL-6. As shown in Fig 3D, depletion of monocytes resulted in markedly reduced IL-1β secretion and near abrogated IL-6 secretion. Thus, monocytes are essential and there is interplay between monocytes and neutrophils in the cytokine response to NDPKs.

## Neither cytokine induction nor preservation of AML cell viability require enzymatic activity of human or pathogen NDPKs

We used site-directed mutagenesis to make a panel of NTP/NDP transphosphorylase and/or protein histidine kinase deficient rNM23-H1 and *E.coli*, *S.pneumoniae*, *C.albicans* rNDPKs and also mutants not able to form higher quaternary structures (tetramers or hexamers) favored by wt NDPKs (summarised in S4 Table). Verification of these mutants is shown in S2 Fig. Despite having variable enzymatic and oligomerization activities, all the mutant proteins retained the ability to elicit IL-1β secretion from Vitamin $D_3$ (VitD$_3$) differentiated THP-1 monocytes (Fig 4A) and the ability to enhance primary AML cell survival (Fig 4B).

## NM23-H1 and pathogen NDPKs activate an NLRP3 inflammasome pathway

IL-1β production by monocytes is intimately associated with activation of the NLRP3 inflammasome which cleaves pro-caspase-1 to release active caspase-1 that in turn cleaves pro-IL-1β, to form active and secreted IL-1β. In many cell types, activation of the NLRP3 inflammasome results in a specialized form of programmed cell death termed pyroptosis [33]. However, in human monocytes lipopolysaccharide (LPS) activates an alternative NLRP3 inflammasome pathway that results in IL-1β secretion in the absence of pyroptosis [34].

We used flow-cytometry to measure caspase-1 activation in normal donor leucocytes following exposure to either rNM23-H1, *E.coli* rNDPK or *S.pneumoniae* rNDPK, using LPS and nigericin (an inducer of NLRP3-dependant pyroptosis [37]) as positive controls. All treatments not only induced caspase-1 activation in monocytes, but also in neutrophils (Fig 5A) which do not bind rNM23-H1 protein (Fig 3B). These findings further suggest that neutrophils become activated downstream of NDPKs binding to monocytes and provide the rationale for the amplification of the IL-1β response seen in TWCs compared to PBMC preparations (Fig 3C) and its loss on depletion of monocytes (Fig 3D).

To further investigate the role of the NLRP3 inflammasome in human monocyte-NDPK responses we used commercially available NLRP3- and caspase-1 deficient (NLRP3def ($N_d$), CASP1def ($C_d$)), and NLRP3 knockout ($N_{ko}$) THP-1 cell lines and NLRP3- and caspase-1 selective inhibitors. We directly assayed caspase-1 activity in differentiated wtTHP-1 and CASP1def THP-1 cells in the presence or absence of nigericin, LPS, rNM23-H1, *E.coli* rNDPK and *C.albicans* rNDPK (Fig 5B). As expected, baseline caspase-1 activity was lower in CASP1def THP-1 monocytes than in untreated wtTHP-1 monocytes (Fig 5B). Nigericin exposure produced an approximately 1000-fold increase in caspase-1 activity in wtTHP-1 monocytes and this response was significantly reduced in CASP1def THP-1 monocytes (Fig 5B). LPS induced a much smaller increase in caspase-1 activity in wtTHP-1 monocytes and again a reduced response in CASP1def THP-1 monocytes was observed. The induction of caspase-1 activity by human (NM23-H1), bacterial (*E.coli*) and yeast (*C.albicans*) rNDPKs in wtTHP-1-

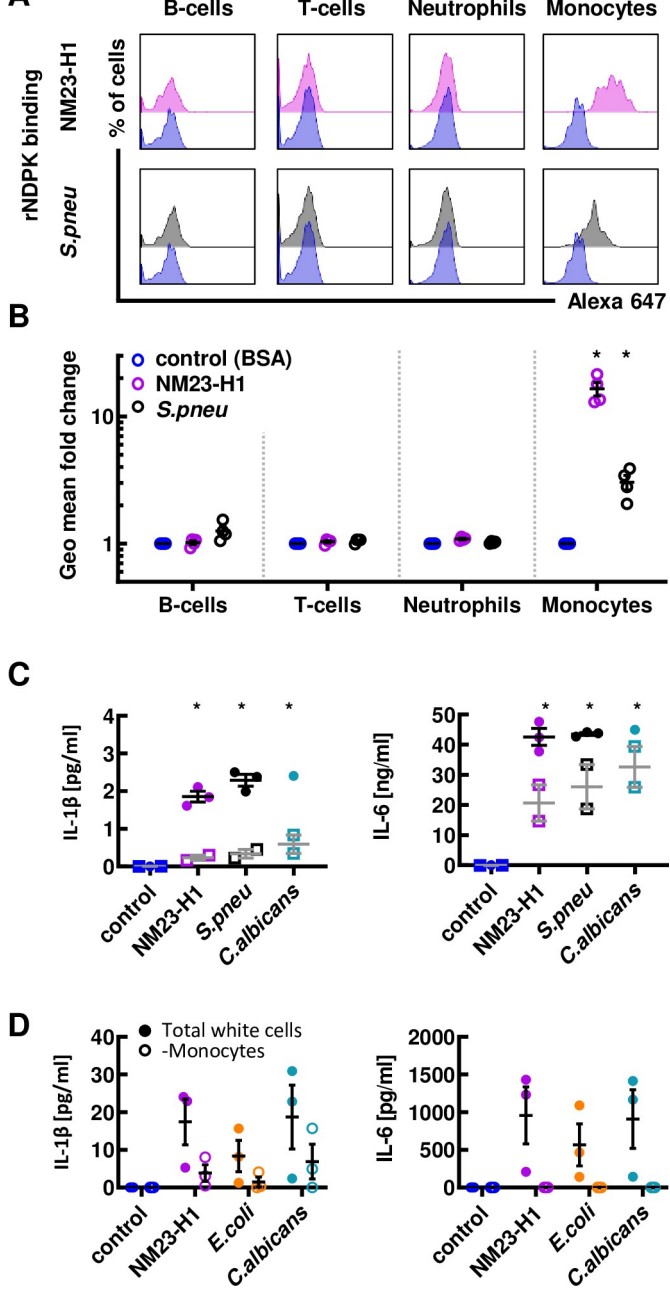

**Fig 3. Promotion of AML cell survival by both human and pathogen rNDPKs is mediated via interaction with non-malignant myeloid cells.** (A) Whole blood from normal donors was diluted and incubated with fluorescently labelled BSA control (in blue), rNM23-H1 (in pink) and S.pneumoniae rNDPKR (in grey). Representative plots for B-cells (CD19), T-cells (CD3), neutrophils (CD11b+CD14-) and monocytes (CD11b+CD14+) are shown (See S2 Fig for gating strategy). (B) Fluorescence geometric mean fold changes compared to BSA control of each individual cell type for n = 4 normal donors. (C) ELISA for IL-1β and IL-6 cytokines in conditioned media (CM) generated incubating for 18 hours diluted whole blood (closed circles) or PBMC (open squares) with rNM23-H1 or rNDPK from S.pneumoniae and C.albicans. (D) Red cells from normal donor's whole blood were lysed and leukocytes were depleted from CD14 + cells (monocytes). Either total white cells or monocyte depleted cells were treated with NM23-H1 or rNDPK for 18h and IL-1β and IL-6 were analyzed by ELISA.

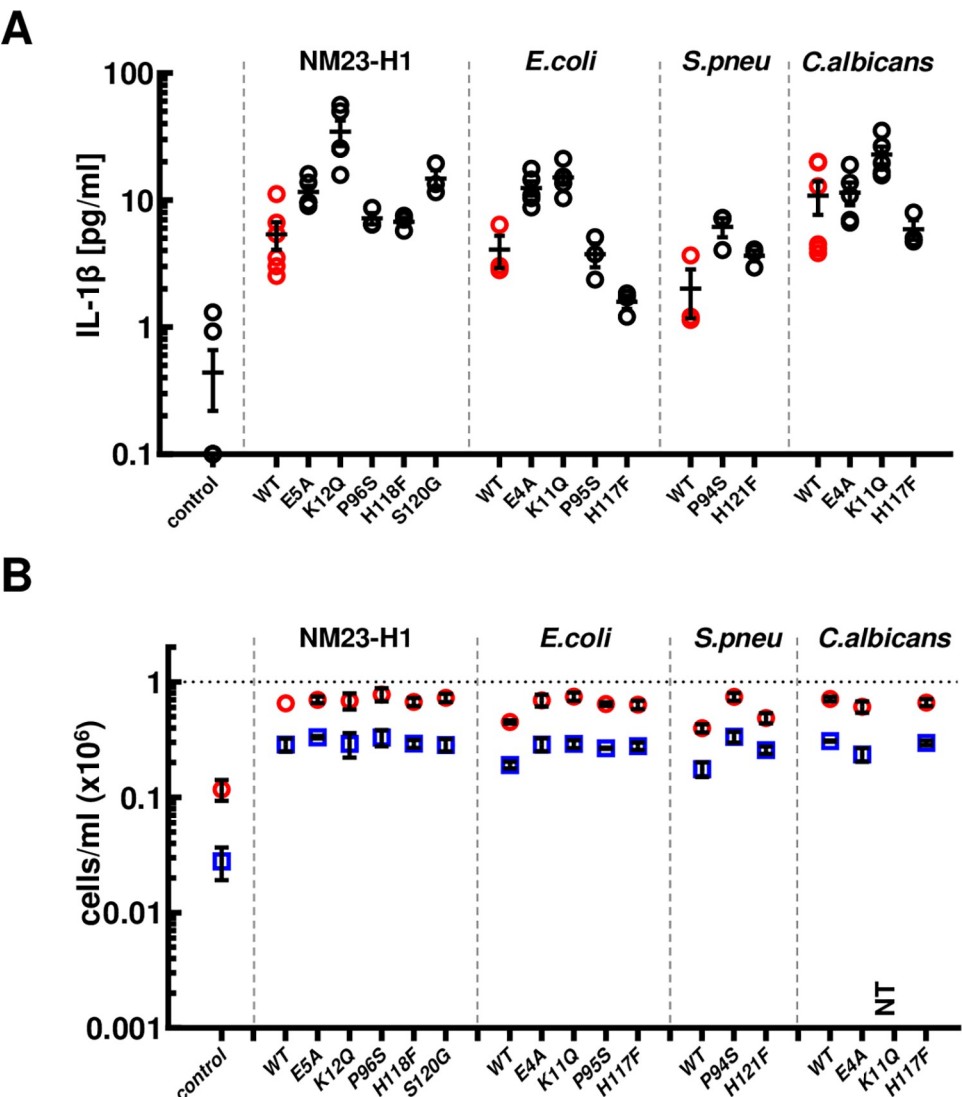

**Fig 4. Induction of cytokines and preservation of AML cell viability does not require enzymatic activity of human and pathogen NDPKs.** (A) Vitamin $D_3$ differentiated THP-1 were incubated for 18 hours in presence of WT (in red) and mutated rNDPK (in black), conditioned media was harvested and IL-1β was measured by ELISA. (B) AML10 was treated for 7 days with WT or mutated rNDPK and cell viability of total cells and blasts was measured by flow cytometry. Dot plots show the mean ± SEM of at least 3 replicates for live cells/ml (red circles) and live blasts/ml (blue squares). All the WT and mutants rNDPK treatments are statistically different from the control; $p < 0.05$, One-way ANOVA with Dunnet's test. Averages are expressed as mean ± SEM.

and CASP1def THP-1- monocytes mirrored the response to LPS, and not the response to nigericin (Fig 5B).

The secretion of IL-1β in response to nigericin was abrogated in CASP1def THP-1 monocytes whereas the response to LPS was not affected (Fig 5C). Secretion of IL-1β in response to human (NM23-H1) bacterial (*E.coli*) and yeast (*C.albicans*) rNDPKs showed an intermediate phenotype and was partially diminished in CASP1def THP-1 monocytes compared to wtTHP-1 monocytes (Fig 5C). The caspase-1 selective inhibitor Ac-YVAD-cmk dramatically reduced IL-1β secretion in response to nigericin, LPS and all three rNDPKs (Fig 5C). Collectively, these

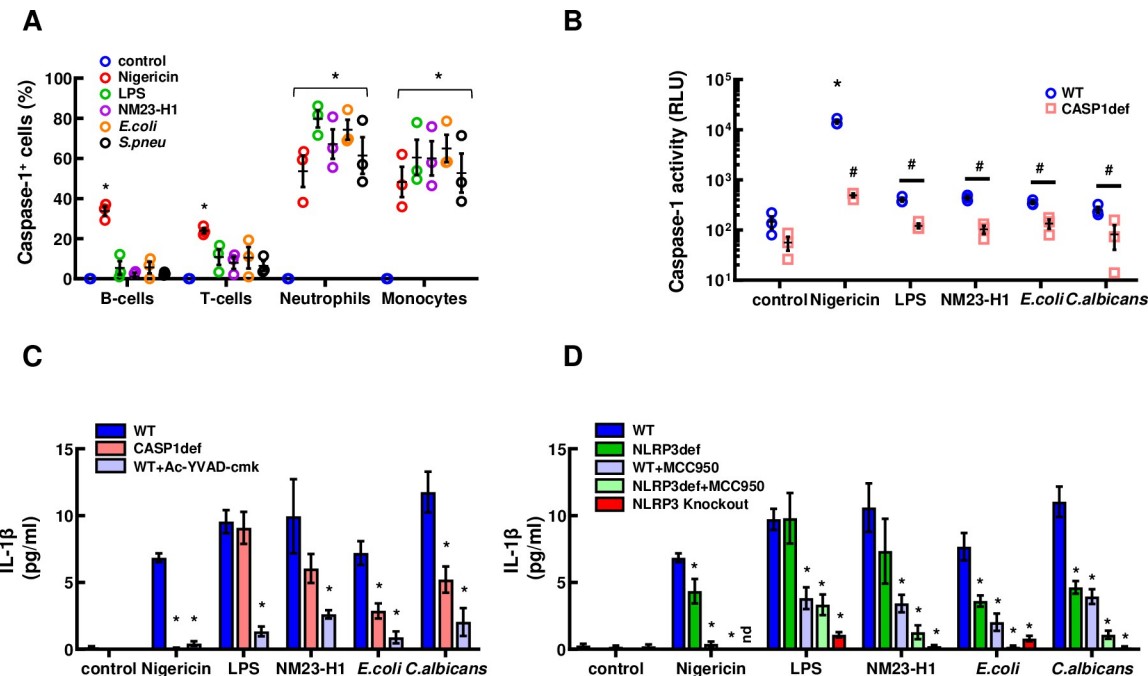

**Fig 5. NM23-H1 and pathogen NDPKs activate the NLRP3 inflammasome pathway.** (A) Whole blood from normal donors was diluted 2:3 and incubated with nigericin, LPS and rNDPKs. The caspase-1 fluorescent dye FLICA 660 was added after 60' and incubation continued for further 90'. Red cells were lysed, and leukocytes were immunophenotyped by flow cytometry. * $p < 0.05$ to controls, One-way ANOVA with Dunnet's test. (B) Caspase-1 specific activity measured with the Caspase-Glo 1 Bioluminescent Inflammasome Assay for both Vitamin $D_3$ differentiated THP-1 wild type (WT, blue circles) and Caspase-1 deficient (CASP1def, pink squares) treated for 150' with nigericin, LPS and rNDPKs. Luminescence of the Ac-YVAD-CHO inhibited reactions was subtracted from the total luminescence. * $p < 0.05$ to controls within same cell line, # $p < 0.05$ between WT and CASP1def; One-way ANOVA with post-hoc Tukey's test. (C) IL-1β ELISA of the conditioned media of Vitamin $D_3$ differentiated THP-1 wild type (WT, in blue), in presence or absence of the Caspase-1 inhibitor Ac-YVAD-cmk (light blue), and of the THP-1 Caspase-1 deficient cell line (pink), treated for 18h with nigericin, LPS or rNDPKs. (D) IL-1β ELISA of the conditioned media from Vitamin $D_3$ differentiated THP-1 wild type (WT, in blue), THP-1 NLRP3 deficient (NLRP3def, in green) and THP-1 NLRP3 knockout (in red), in presence or absence of the NLRP3 inhibitor MCC950 (light blue and green), treated for 18h with nigericin, LPS or rNDPKs. Averages are expressed as mean ± SEM. * $p < 0.05$ to controls, One-way ANOVA with Dunnet's test.

findings indicate that IL-1β secretion by monocytes in response to NM23-H1 and pathogen rNDPKs is caspase-1 dependent.

VitD$_3$-differentiated NLRP3def-THP-1 demonstrated variably reduced IL-1β secretion in response to nigericin, rNM23-H1, *E.coli* rNDPK and *C.albicans* rNDPK whereas the response to LPS appeared unchanged (Fig 5D). Treatment of both wtTHP-1 and NLRP3def-THP-1 monocytes with the NLRP3 inhibitor MCC950 reduced LPS-induced IL-1β secretion (Fig 5D). Similarly, MCC950 markedly diminished IL-1β secretion by wtTHP-1 monocytes in response to nigericin, rNM23-H1, *E.coli* rNDPK and *C.albicans* rNDPK (Fig 5D). In addition, MCC950 further diminished IL-1β secretion by NLRP3def-THP-1 monocytes in response to these stimuli indicating that rNDPK induced IL-1β secretion by monocytes is also NLRP3 dependent. Importantly, IL-1β production, was completely lost in N$_{ko}$ THP-1 cells (Fig 5D).

## NM23-H1 and pathogen NDPK activation of the NLRP3 inflammasome pathway occurs without pyroptosome formation

Active inflammasomes are complexes composed of multiple copies of NLRP3, ASC and pro-caspase-1 proteins. In the classical NLRP3 pathway they ultimately form a single very large aggregate termed the pyroptosome (sometimes referred to ASC specks) which coincide with

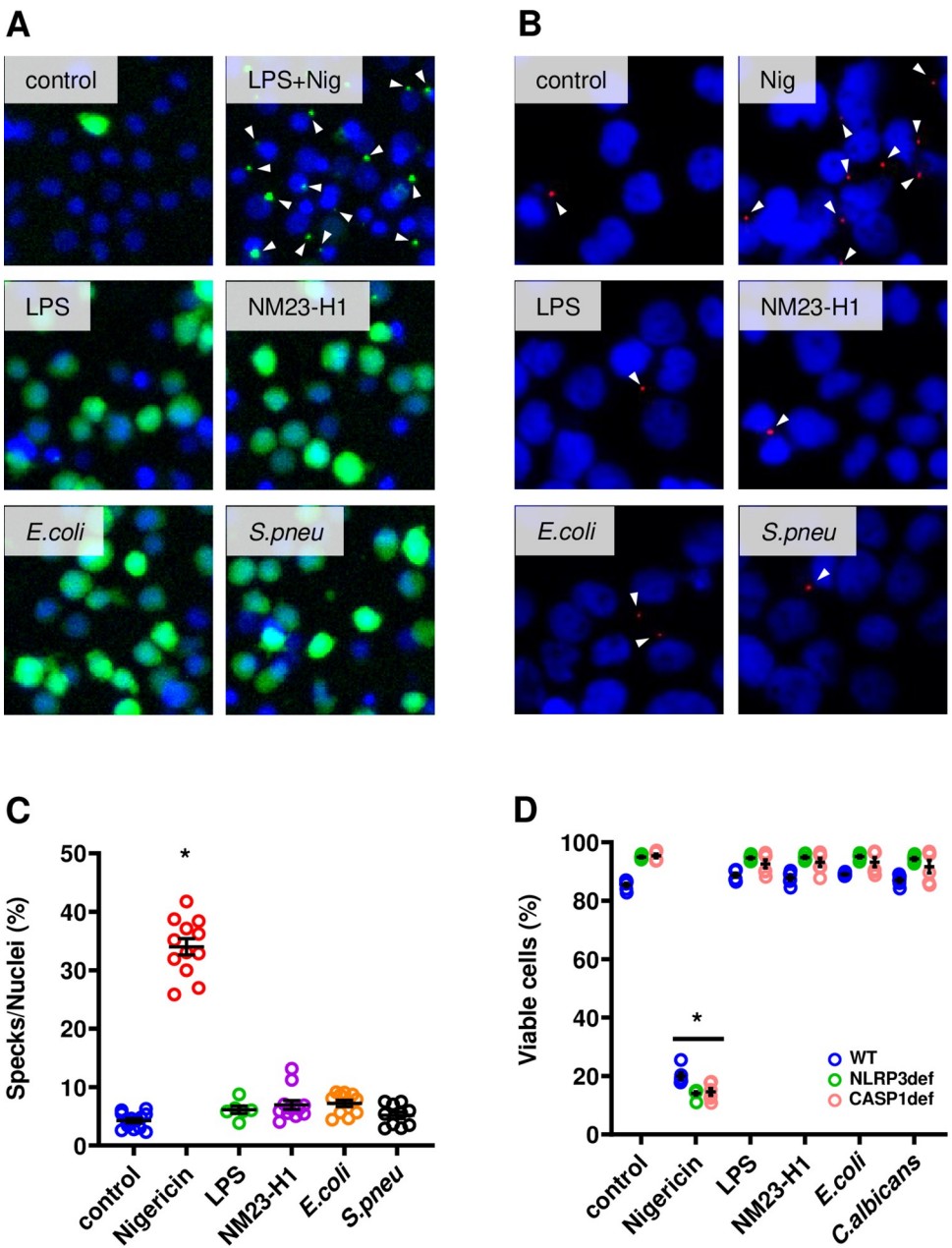

**Fig 6. NM23-H1 and pathogen NDPKs activation of the NLRP3 inflammasome pathway occurs without pyroptosome formation.** (A) Vitamin D3 differentiated THP-1 ASC::GFP cells were treated for 6 hours with LPS and rNDPKs or for 3 hours with LPS to induce ASC::GFP protein with subsequent treatment with Nigericin for further 3 hours (LPS+Nig). ASC::GFP protein was visualized by immunofluorescent microscopy. (B) Immunofluorescent staining of ASC specks in Vitamin D3 differentiated wtTHP-1. Arrows indicate ASC specks. (C) Dot plot of the quantification of the number of ASC specks per total nuclei in THP-1 wild type. * p<0.05 compared to control, One-way ANOVA with Dunnet's test within the same cell line. (D) Dot plot of the viability of Vitamin D3 differentiated THP-1 WT (WT, in blue), NLRP3 deficient (NLRP3def, in green) and Caspase-1 deficient (CASP1def, in pink) after treatment with nigericin, LPS, and rNDPKs. Viability was measured by flow cytometry and the Forward Scatter/Side Scatter parameters. * p<0.05, One-way ANOVA with Dunnet's test within the same cell line. Averages are expressed as mean ± SEM.

death by pyroptosis [33]. In the more recently described alternative NLRP3-inflammasome pathway, IL-1β secretion by human monocytes occurred in the absence of pyroptosome formation and without pyroptotic cell death [34].

We used VitD$_3$-differentiated THP-1 cells stably expressing an NF-κB inducible ASC::GFP fusion gene (ASC::GFP THP-1) to determine whether rNM23-H1 and pathogen rNDPK responses are associated with pyroptosome formation. ASC::GFP expression was induced after 3–6 hours exposure to LPS, rNM23-H1, *E.coli* rNDPK and *S.pnuemoniae* rNDPK (Figs 6A and S3) demonstrating that like LPS, human and pathogen NDPKs activate NF-κB in human monocytes. However, nigericin induces pyroptosis without NF-κB activation. Thus as expected, nigericin treatment alone did not induce ASC::GFP expression (S3 Fig). However, pyroptosome formation was observed when nigericin was used to treat LPS exposed (3hr) ASC::GFP expressing cells (Figs 6A and S3). To investigate more closely the induction of pyroptosomes by nigericin in our system, we used ASC-immunofluorescence in wtTHP-1 monocytes. We observed pyroptosome-specks in response to nigericin alone without co-treatment with LPS (Fig 6B and 6C). However, we again failed to detect increased pyroptosome formation in response to either LPS or rNM23-H1 and rNDPKs (Fig 6B and 6C). Consistent with the formation of pryoptosomes, nigericin treatment but not LPS, rNM23-H1, *E.coli* rNDPK and *S.pnuemoniae* rNDPK induced rapid cell death of THP-1 monocytes (Fig 6D).

## Activation of the NLRP3 inflammasome pathway by NM23-H1 and pathogen NDPKs is independent of TLR4 signaling

The production of recombinant proteins in *E.coli* risks contamination with LPS. Although all our experiments using rNDPKs were performed using LPS-neutralizing polymyxin-B (PMB), it was important to rule out the possibility of inflammasome activation by rNDPKS due to low-level LPS contamination. We therefore exploited Dual Reporter THP-1 cells (THP1-Dual) and THP1-Dual TLR4 KO cells [38]. These cells have inducible luciferase and SEAP reporter genes, allowing the concomitant study of activation of the IRF and NF-κB pathways, respectively. In addition, THP1-Dual TLR4 KO cells express a truncated TLR4 unable to respond to LPS. For these experiments we also produced human and pathogen rNDPKs in endotoxin-free ClearColi BL 21 cells that contain genetically modified LPS that does not trigger an endotoxic response in humans [39]. As in previous experiments we used LPS as a positive control.

As expected, LPS induced NF-κB and IRF activation and secretion of IL-1β in TLR4-competent THP1-Dual monocytes. Responses which were markedly diminished or abrogated by PMB and the TLR4-inhibitor TAK-242 (resatorvid) [40]. Also as expected, LPS did not evoke NF-κB and IRF activation or secretion of IL-1β in THP1-Dual TLR4 KO monocytes. In contrast, ClearColi derived rNM23-H1 and pathogen rNDPKs induced NF-κB, IRF and IL-1β responses in both THP1-Dual and THP1-Dual TLR4 KO monocytes when even in the presence of PMB or TAK-242 (Fig 7A and 7C) ruling out that these actions were mediated by contaminating LPS. The activation of NF-κB reporter gene expression by NDPKs in THP1-Dual and THP1-Dual TLR4 KO monocytes is consistent with NDPK induction of the NF-κB inducible ASC::GFP fusion gene in ASC::GFP THP-1 monocytes (Figs 6A & S5). Whereas induction of the NF-κB reporter was consistent across the NDPKS tested (Fig 7A), the response of the IRF reporter and secretion of IL-1β were more variable amongst the tested NDPKs (Fig 7B and 7C) and correlated with each other (Fig 7D).

## Discussion

The demonstration here that human, bacterial and fungal derived NDPKs converge on signaling via a common TLR4-independent NLRP3 inflammasome pathway, has implications for

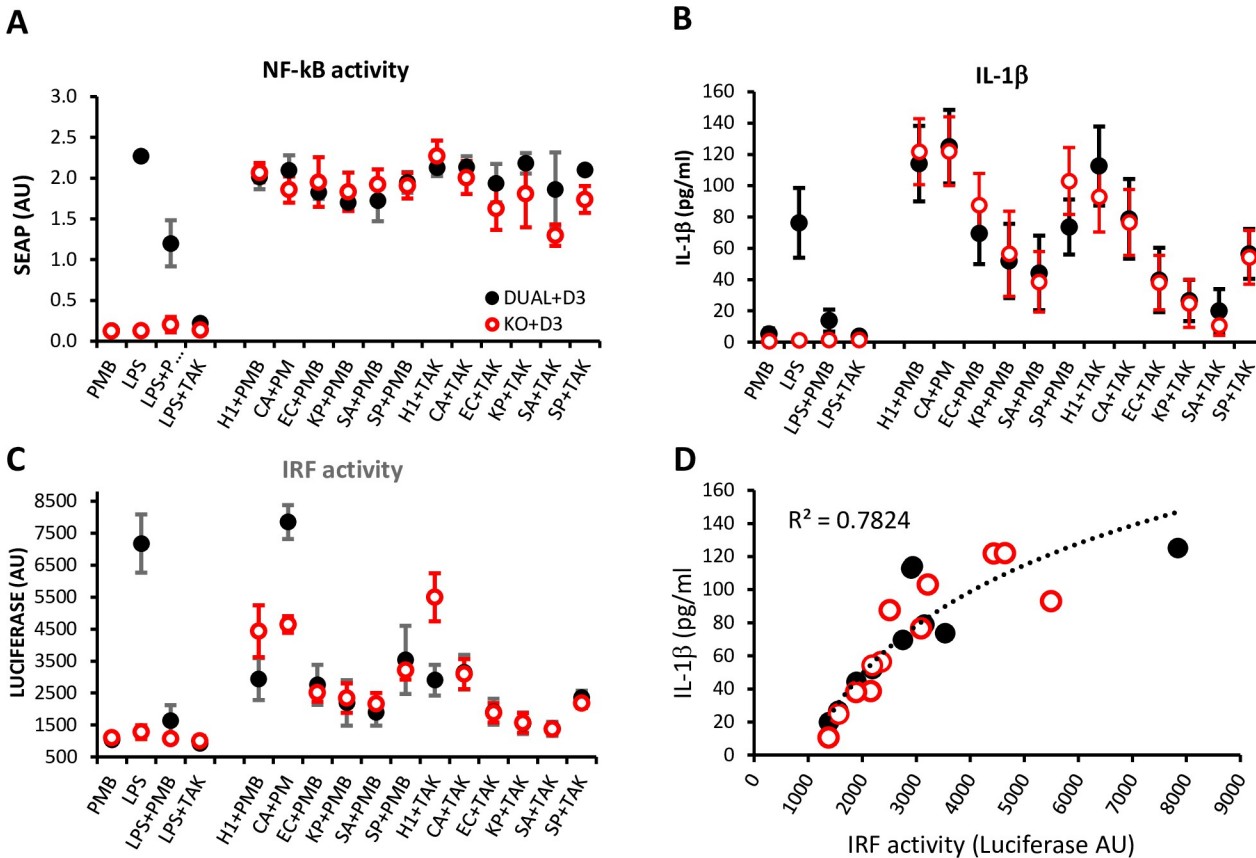

**Fig 7. NDPK induced Activation of NF-κB, secretion of IL-1β and IRF activation occur independently of TLR4 signaling.** THP1-Dual (black closed circles) and THP1-Dual TLR4 KO (red open circles) were differentiated into monocytes with Vitamin D3. Following 18 hours exposure to the treatments shown, the following were measured: (A) NF-κB-SEAP reporter gene activity. (B) IL-1β ELISA of the conditioned media, and (C) IRF-Lucia-Luciferase reporter gene activity. (D) Data for IRF Lucia-Luciferase reporter gene activity were plotted against IL-1β ELISA results for Vitamin D3-differentiated THP1-Dual (black) and THP1-Dual TLR4 KO (red) monocytes. Line of fit is log[10]. PMB; polymyxin-B. TAK; TAK-242. Data are the mean of n = 4–6 ± SEM.

the role of infections in AML and MDS progression. Although a novel concept in these diseases, the association between microbiota and solid cancers is becoming accepted [41–45] and the association of inflammation with cancer is well established [44,46]. In addition, the association of sterile NLRP3 inflammasome activation with AML and MDS is also well accepted [10–16]. To date, the role of NLRP3 inflammasomes in myeloid malignancies has focused on classical pyroptotic pathways, where cells dying by pyroptosis release DAMPS that further activate inflammation via TLRs including TLR4 [13,15,17,47–49]. However, our study has highlighted the potential of pathogen derived NDPKs acting as PAMPS to activate NLRP3 independently of either TLR4 or pyroptosis. Therefore, our findings widen the potential impact of NLRP3 signaling processes in these settings.

Of note is that many studies of PAMP-associated inflammation utilise LPS as model. However, LPS is restricted to Gram-negative bacterial infections. Here we show that NDPKs from Gram-positive (*S.aureus*, *S.pnuemoniae*) and Gram-negative (*E.coli*, *K.pneumoniae*) bacteria and at least one fungus (*C.albicans*), activate inflammatory mechanisms and promote AML cell survival. Furthermore, we demonstrate that although LPS and NDPKs mediate inflammation via NLRP3 and caspase 1 activation, the perception of these distinct classes of PAMP by monocytes is different with LPS being detected by TLR4 and NDPKs not.

AS seen in our previous studies and those of others the survival response of AML samples to NDPKs was selective. We have not observed the positive or negative responses to be associated with any available clinical variable such as blast count or differentiation status of AML blasts. One explanation for the differential responses displayed by AML cells are differential responses to downstream consequences of inflammasome activation. For example, an ex vivo functional screen of 94 cytokines, identified that IL-1β elicited expansion of myeloid progenitors in some but not all AMLs [7].

A notable finding was the correlation between variable NDPK-induced IL-1β secretion and IRF reporter gene activation in both THP1-Dual and THP1-Dual TLR4 KO monocytes. It is not possible from our data to determine the relationship between IRF signaling and monocyte IL-1β secretion.

NM23-H1 was amongst the strongest stimulators of IL-1β secretion and IRF activation whereas *S.pneumoniae* NDPK elicited lower responses. This may reflect differential binding characteristics of different NDKs to human monocytes since we also observed greater binding (~10 fold) of fluorescently labelled NM23-H1 than *S.pneumoniae* NDPK to healthy donor monocytes.

Although our studies implicate pathogen NDPKs in inflammasome activation, additional studies are required. One question that remains is how monocytes perceive the presence of NDPKs. Future studies should focus on what might be the receptor for NDPKs. Furthermore, although our serological analyses support the argument that humans are exposed to pathogen NDPKs *in vivo*, their role for disease progression *in vivo* and how this is impacted by the presence B1 IgM antibodies now requires investigation.

## Supporting information

**S1 Fig. Gating strategy for flow cytometry of AML samples.** Two AML samples are shown as an example of the gating strategy. Live cells were identified by Forward Scatter/Side Scatter plot; blast cells were then identified by size and their main blast marker at diagnosis (either CD34 (A) or CD117 (B)). Finally, dual positivity for the second blast marker (either CD117 (A) or CD34 (B)) was verified with a CD117/CD34 plot.
(TIF)

**S2 Fig. Gating strategy for analyzing binding of Alexa 647 labelled protein.** Left: Representative forward and side scatter plot of red-cell lysis prepared donor blood showing live gate. Centre: Neutrophils were identified as CD11b+CD14- and monocytes as CD11b+CD14+. Right: in a separate tube, T cells were identified as CD3+CD19- and B-cells as CD3+CD19-.
(TIF)

**S3 Fig. rNDPK mutants kinase activity validation.** To create mutants all mutagenesis reactions were performed directly on the NDPKs sequences cloned in the pET15b backbone. Alignment amongst NM23-H1 and bacterial and fungal NDPK was performed with Clustal 2.1. This identified the conserved residues and allowed to infer, based on NM23-H1 literature, which of the residues would have impaired the enzyme function and structural organization on pathogen NDPK. Primers were designed with PrimerX online software (https://www.bioinformatics.org/primerx/) with design protocol for QuikChange Site-Directed Mutagenesis and codon optimized for *E.coli* expression systems. Mutagenesis reaction was performed using QuikChange II Site-Directed Mutagenesis kit (Agilent Technologies) following the manufacturer's instructions. Mutations were confirmed by sequencing and the plasmid were transformed in BL21(DE3) for protein production. (A) Recombinant NDPKs, wild type (WT) and mutants, were pre-incubated in the presence or absence of ATP and run on two SDS-PAGE.

One was stained with Sypro Ruby (SR) protein stain to detect total protein levels (on top) and the other was transferred and immunoblotted for an anti-N1-phosphohistidine antibody (αP-His). The autophosphorylation of rNDPKs was analyzed with ATP as substrate in the presence of MgCl$_2$. 50ng of stock rNDPKs were incubated with and without 1mM ATP for 5 minutes at room temperature in 30μl buffer (20mM Tris-HCl, pH 8.0; 150mM NaCl and 1mM MgCl$_2$). Reaction was stopped by adding 5mM EDTA for Mg$^{2+}$ chelation. 10μl of 4X loading buffer (50% Glycerol; 10% β-Mercaptoethanol; 7.5% SDS; 300 mM Tris/HCl; pH 6.8; 0.25% Bromophenol blue) were added and proteins separated without heat denaturation on two 15% SDS-PAGE. The first gel was stained with SYPRO Ruby Protein Gel Stain (ThermoFisher Scientific) according to the manufacturer's instructions. Proteins from the second gel were transferred onto a 0.2μm nitrocellulose membrane (Amersham Protran) and probed overnight at 4°C with 1:1000 αN1-Phosphohistidine (1-pHis) antibody (clone SC1-1, Merck Millipore), followed by 1:80,000 horse radish peroxidase conjugated secondary antibody (ThermoFisher Scientific) at room temperature for 60 minutes. The blots were developed using SuperSignal West Femto Maximum Sensitivity Substrate (ThermoFisher Scientific) and visualized with the Vilber Fusion FX. (B) Relative NDP-kinase activity of WT (in red) and mutant NDPK (in black). The transphosphorylase activity of the NDPKs was measured in a reaction mixture containing 50mM Tris-HCl, pH 7.5, 2mM MgCl$_2$, 1mM DTT and 0.01% BSA (Buffer A). Stock proteins were diluted in Buffer A to a concentration of 300pM and 1 volume of the rNDPK solution was mixed with 1 volume of substrate mixture (200μM GTP and 20μM ADP in Buffer A) in a 384 well plate. The mixture was incubated at room temperature for 30 minutes and 2 volumes of the Kinase-GLO reagent (Promega, V6711), containing an ATP dependent firefly luciferase, were added and luminescence measured using a plate reader (PerkinElmer-EnVision). Six different concentrations of ATP (0–10 μM) were used as standards for a calibration curve to calculate the amount of ATP produced by the different NDPKs. All readings shown were obtained under conditions where the ATP formation was still linear with time and enzyme concentration.
(TIF)

**S4 Fig. NfκB dependent ASC::GFP fusion gene in THP-1 ASC::GFP cell line is induced by LPS and rNDPKs.** THP-1 ASC::GFP were treated for 3 and 6 hours with Nigericin 20μM, LPS and rNDPK 2μg/ml. In LPS+Nig treatment, was first induced for 3h with LPS 2μg/ml and then Nigericin 20μM was added to induce speck formation. T0 = untreated cells at time 0; T3 = 3 hours; T6 = 6 hours.
(TIF)

**S5 Fig. Uncropped unadjusted images for gels shown in S3 Fig part A.**
(TIF)

**S1 Table. Percent Identity Matrix of NDPK Protein sequences.** NM23-H1 (H1), NM23-H2 (H2), *Escherichia coli* (EC), *Pseudomonas aeruginosa* (PA), *Staphylococcus aureus* (SA), *Staphylococcus pneumoniae* (SP), *Klebsiella pneumoniae* (KP), *Candida albicans* (CA), *Candida glabrata* (CG) were aligned and Percent Identity Matrix was generated with Clustal 2.1.
(TIF)

**S2 Table.** Assay of humoral IgG (A) and IgM (B) reactivity against human and pathogen NDPKs. Table shows Individual data used to plot graphs in Fig 1C & 1D. AMLs were selected and ranked with a range of CRP (C-reactive protein) levels (mg/L). ND = normal donor; nt = not tested. As shown in the table all samples tested for IgM levels were also tested for IgG levels.
(TIF)

**S3 Table. Viable cells and blasts after 7 days of treatment.** Primary AML cells were plated out at 1x106 cells/ml and treated with rNDPKs for 7 days. Viable total cells (A) and blasts (B) were enumerated by Flow cytometry and the use of counting beads and are presented in the Table. H1: rNM23-H1, EC: rNDPK E.coli; SP: rNDPK S.pneumoniae; KP: rNDPK K.pneumoniae; SA: rNDPK S.aureus; CA: rNDPK C.albicans. nt = not tested.
(TIF)

**S4 Table. NDPK mutants and effects.** NM23-H1 mutants and their effects. Green indicates retained function, yellow a moderate loss while red impaired activity. H: Hexamer; D: Dimer; M: Mix of dimeric and hexameric structures. E5A Glu-5 to Ala; K12Q Lys-12 to Gln; H118F His-118 to Phe, P96S Pro-96 to Ser, S120G Ser-120 to Gly.
(TIF)

# Acknowledgments

We are grateful to Dr Adrian Shields for critical review of the manuscript.

# Author Contributions

**Conceptualization:** Thomas Wieland, Mark T. Drayson, Farhat L. Khanim, Christopher M. Bunce.

**Data curation:** Daniel Wiseman, Deepti P. Wilks, Christopher M. Bunce.

**Formal analysis:** Sandro Trova, Santosh Lomada, Thomas Wieland, Farhat L. Khanim, Christopher M. Bunce.

**Funding acquisition:** Farhat L. Khanim, Christopher M. Bunce.

**Investigation:** Sandro Trova, Fei Lin, Santosh Lomada, Matthew Fenton, Bhavini Chauhan, Alexandra Adams, Avani Puri, Alessandro Di Maio, Daniel Sewell, Kirstin Dick, Margaret Goodall, Farhat L. Khanim.

**Methodology:** Sandro Trova, Santosh Lomada, Thomas Wieland, Margaret Goodall.

**Project administration:** Deepti P. Wilks.

**Resources:** Daniel Wiseman.

**Supervision:** Thomas Wieland, Daniel Wiseman, Mark T. Drayson, Farhat L. Khanim, Christopher M. Bunce.

**Validation:** Sandro Trova, Christopher M. Bunce.

**Writing – original draft:** Christopher M. Bunce.

**Writing – review & editing:** Sandro Trova, Daniel Wiseman, Mark T. Drayson, Farhat L. Khanim.

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
