## [Decision Letter · Decision Letter 0]

13 Apr 2023

PONE-D-23-03267Pathogen and human NDPK-proteins promote AML cell survival via monocyte NLRP3-inflammasome activation.PLOS ONE

Dear Dr. Bunce,

Thank you for submitting your manuscript to PLOS ONE. After careful consideration, we feel that it has merit but does not fully meet PLOS ONE’s publication criteria as it currently stands. Therefore, we invite you to submit a revised version of the manuscript that addresses the points raised during the review process.

We look forward to receiving your revised manuscript.

Kind regards,

Francesco Bertolini, MD, PhD

Academic Editor

PLOS ONE

“This research was supported by a grant from Blood Cancer UK (Grant 17011). SL has been supported by the state of Baden-Württemberg within the PharmCompNet project.  We are grateful to Dr Adrian Shields for critical review of the manuscript.”

“This research was supported by a grant from Blood Cancer UK (Trading name of Bloodwise) Grant 17011) CB, FK (https://bloodcancer.org.uk/). SL has been supported by the state of Baden-Württemberg within the PharmCompNet project. The funders had no role in study design, data collection and analysis, decision to publish, or preparation of the manuscript.”

4. Please include your tables as part of your main manuscript and remove the individual files. Please note that supplementary tables (should remain/ be uploaded) as separate "supporting information" files

Additional Editor Comments:

Dear Authors, thank you for submitting this interesting manuscript to PLOSone. Please answer to the comments from the 3 Reviewers, experts in the field.

Reviewers' comments:

Reviewer's Responses to Questions

**Comments to the Author**

1. Is the manuscript technically sound, and do the data support the conclusions?

Reviewer #1: Yes

Reviewer #2: Yes

Reviewer #3: Yes

2. Has the statistical analysis been performed appropriately and rigorously? 

Reviewer #1: Yes

Reviewer #2: Yes

Reviewer #3: Yes

3. Have the authors made all data underlying the findings in their manuscript fully available?

Reviewer #1: Yes

Reviewer #2: Yes

Reviewer #3: Yes

4. Is the manuscript presented in an intelligible fashion and written in standard English?

Reviewer #1: Yes

Reviewer #2: Yes

Reviewer #3: Yes

5. Review Comments to the Author

Reviewer #1: Acute myeloid leukaemia (AML) and myelodysplastic syndromes MDS) are blood cancers that are associated with frequent infections. Experimental evidences are reported here that infections may promote or drive cancer progression.

Th authors show that nucleoside diphosphate kinases (NDPK) from both Gram-positive (S. aureus, S. pneumoniae) and Gram-negative (E. coli, K. pneumoniae) bacteria and fungi (C. albicans), activate inflammatory mechanisms and promote AML cell survival.

NDPKs from pathogens have been shown to recapitulate the pro-survival effect of human NM23-H1 protein on AML cells, where NM23-H1 shows significant sequence similarity with NDPKs from pathogens.

These findings widen the understanding of the physio-pathological relationships likely existing between infections and tumor growth and progression.

The only concern that can rise at this stage is to make the Results and the Discussion more focused and better defined.

Reviewer #2: The paper from Trova S. et al. identifies a new role for pathogen NDPK-proteins in promoting survival of AML blasts through induction of an inflammatory response. These data raise the important implication that pathogen infections in AML patients could play an active role in the leukemogenic process, rather than being simply a consequence. I think that the paper is technically sound and clearly written. The experiments were conducted appropriately and the authors discuss their data in agreement with their findings. However, since the inflammatory response induced by NDPK-proteins seems to be mediated mainly by normal monocytes, how can the authors explain that around 60% (9/16) of AML tested samples are not responders and do not show any survival advantage? Is there any correlation between response and previous pathogen infections of the AML patients from which blasts were derived, and/or the level of differentiation of the AML blasts, and/or any other clinical feature of AML?

I also have a minor technical issue. In supplementary Figure S2, the blots showing total protein levels of the different mutants by Sypro Ruby staining are not visible. It is difficult to judge equal protein expression and loading.

Reviewer #3: Trova and colleagues describe in this manuscript the show that NDPKs from different strains of bacteria and one fungus are able to activate inflammatory mechanisms and promote AML cell survival in patients. They showed that the activation is mediated by monocyte and in cascade by neutrophils. Moreover, they showed that the release of IL-1beta depends on the activity of caspase-1.

The manuscript is well written and all the experiments are logical and well described.

However this reviewer has some issues on the experiments:

Major revision:

1. In figure 1C and 1D the authors measured the IgM and IgG levels in different AMLs and healthy donors. They used for the IgM 14 AMLs and 2 Healthy donors, while for the IgG 45 AMLs and 3 Healthy donors. It is not clear why there is this discrepancy in the numbers between the two experiments, and why the authors did not use the same sample for IgG and IgM. The study would have been more interesting if within the same sample both IgG and IgM were measured

2. In Figure 2A FACs plot of cells are presented. Authors discriminates alive cells (red) and dead cells (black) by their morphology. This analysis seems to be too much arbitrary. It would have been better to use a marker of cell vitality (7-AAD, DAPI, etc.) and repeat the analysis. Moreover, it is not clear how the authors have counted the alive cells for the graph 2B.

3. It would be useful in experiments showed in Figure 3A, to show how cells are subdivided in B cells, T cells, monocytes and neutrophils. A suggestion might be showing FACS plot subdividing firstly the different subpopulations and than with the positivity towards fluorescent NDPK

4. It would be useful to show FACS plot of monocytes depleted samples in Figure 3D, to demonstrate the purity of isolated monocytes

5. In Figure 5 authors claimed to have used THP1 deficient or knockout for caspase1. It would be useful to show a western blot indicating the lack of the protein.

Minor comments:

1. Table 1 should be put in order after Figure 1

2. Immunofluorescence images are completely out of focus. They should be acquired better

6. PLOS authors have the option to publish the peer review history of their article (what does this mean?). If published, this will include your full peer review and any attached files.

Reviewer #1: No

Reviewer #2: No

Reviewer #3: No

---

## [Author Response · Author response to Decision Letter 0]

2 Jun 2023

a response to editor and reviewers has been uploaded

---

## [Decision Letter · Decision Letter 1]

20 Jun 2023

Pathogen and human NDPK-proteins promote AML cell survival via monocyte NLRP3-inflammasome activation.

PONE-D-23-03267R1

Dear Dr. Bunce,

We’re pleased to inform you that your manuscript has been judged scientifically suitable for publication and will be formally accepted for publication once it meets all outstanding technical requirements.

Kind regards,

Francesco Bertolini, MD, PhD

Academic Editor

PLOS ONE

Additional Editor Comments (optional):

Reviewers' comments:

Reviewer's Responses to Questions

**Comments to the Author**

1. If the authors have adequately addressed your comments raised in a previous round of review and you feel that this manuscript is now acceptable for publication, you may indicate that here to bypass the “Comments to the Author” section, enter your conflict of interest statement in the “Confidential to Editor” section, and submit your "Accept" recommendation.

Reviewer #1: All comments have been addressed

Reviewer #2: All comments have been addressed

Reviewer #3: All comments have been addressed

2. Is the manuscript technically sound, and do the data support the conclusions?

Reviewer #1: (No Response)

Reviewer #2: (No Response)

Reviewer #3: Yes

3. Has the statistical analysis been performed appropriately and rigorously? 

Reviewer #1: (No Response)

Reviewer #2: (No Response)

Reviewer #3: Yes

4. Have the authors made all data underlying the findings in their manuscript fully available?

Reviewer #1: (No Response)

Reviewer #2: (No Response)

Reviewer #3: Yes

5. Is the manuscript presented in an intelligible fashion and written in standard English?

Reviewer #1: (No Response)

Reviewer #2: (No Response)

Reviewer #3: Yes

6. Review Comments to the Author

Reviewer #1: (No Response)

Reviewer #2: (No Response)

Reviewer #3: All the comments of this reviewer were correctly addressed. This reviewer has no further comments to be discussed

7. PLOS authors have the option to publish the peer review history of their article (what does this mean?). If published, this will include your full peer review and any attached files.

Reviewer #1: No

Reviewer #2: No

Reviewer #3: No

---

## [Editor Report · Acceptance letter]

26 Jun 2023

PONE-D-23-03267R1 

Pathogen and human NDPK-proteins promote AML cell survival via monocyte NLRP3-inflammasome activation. 

Dear Dr. Bunce:

I'm pleased to inform you that your manuscript has been deemed suitable for publication in PLOS ONE. Congratulations! Your manuscript is now with our production department. 

Kind regards, 

on behalf of

Dr. Francesco Bertolini 

Academic Editor

PLOS ONE